# A Dual-Channel MoS_2_-Based Selective Gas Sensor for Volatile Organic Compounds

**DOI:** 10.3390/nano14070633

**Published:** 2024-04-05

**Authors:** Esra Kuş, Gülay Altındemir, Yusuf Kerem Bostan, Cihat Taşaltın, Ayse Erol, Yue Wang, Fahrettin Sarcan

**Affiliations:** 1Department of Physics, Faculty of Science, Istanbul University, Vezneciler, Istanbul 34134, Turkey; esra.kus@ogr.iuc.edu.tr (E.K.); yusufkerembostan@gmail.com (Y.K.B.); ayseerol@istanbul.edu.tr (A.E.); 2Materials Institute, TUBITAK Marmara Research Center, Gebze, Kocaeli 41470, Turkey; gulayaltndmr@gmail.com (G.A.); cihat.tasaltin@tubitak.gov.tr (C.T.); 3Department of Physics, School of Physics, Engineering and Technology, University of York, York YO10 5DD, UK

**Keywords:** 2D materials, TMDs, MoS_2_, Gas sensor, volatile organic compounds, VOC, surface functionalization, UV ozone

## Abstract

Significant progress has been made in two-dimensional material-based sensing devices over the past decade. Organic vapor sensors, particularly those using graphene and transition metal dichalcogenides as key components, have demonstrated excellent sensitivity. These sensors are highly active because all the atoms in the ultra-thin layers are exposed to volatile compounds. However, their selectivity needs improvement. We propose a novel gas-sensing device that addresses this challenge. It consists of two side-by-side sensors fabricated from the same active material, few-layer molybdenum disulfide (MoS₂), for detecting volatile organic compounds like alcohol, acetone, and toluene. To create a dual-channel sensor, we introduce a simple step into the conventional 2D material sensor fabrication process. This step involves treating one-half of the few-layer MoS₂ using ultraviolet–ozone (UV-O_3_) treatment. The responses of pristine few-layer MoS₂ sensors to 3000 ppm of ethanol, acetone, and toluene gases are 18%, 3.5%, and 49%, respectively. The UV-O_3_-treated few-layer MoS₂-based sensors show responses of 13.4%, 3.1%, and 6.7%, respectively. This dual-channel sensing device demonstrates a 7-fold improvement in selectivity for toluene gas against ethanol and acetone. Our work sheds light on understanding surface processes and interaction mechanisms at the interface between transition metal dichalcogenides and volatile organic compounds, leading to enhanced sensitivity and selectivity.

## 1. Introduction

Environmental pollution, caused in part by the Industrial Revolution, continues to have a detrimental impact on our planet. One of the significant pollution sources is volatile organic compounds (VOCs), which are produced by various human activities from paint manufacturing to pharmaceutical industries and from hydrocarbon fuel burning to refrigerants [1]. VOCs also have direct negative health effects on humans, such as eyes, nose, and throat irritation, respiratory problems, nausea, and damage to the central nervous system and other organs [2,3]. It is also reported that these gases can cause cancer [4]. VOCs can cause ground-level ozone formation through atmospheric photochemical reactions with NO_x_, harming other living organisms and the environment [5]. Due to these significant concerns, it is critical to measure and monitor VOCs both indoors and outdoors.

In order to detect the VOCs, there are many existing III-V or II-VI group semiconductor- [6] and polymer-based gas sensors [7], with the active materials being copper(II) oxide (CuO) [8], copper(I) oxide (Cu_2_O) [9], zinc oxide (ZnO) [10], titanium dioxide (TiO_2_) [11], tin (IV) oxide (SnO_2_) [12], and indium oxide (In_2_O_3_) [13]. Beyond these materials, two-dimensional (2D) materials have emerged as promising candidates for sensing applications in recent years due to their high surface-to-volume ratio and the presence of dangling bonds on their surfaces and edges, making them ideal materials for physisorption- and chemisorption-based sensors. Apart from graphene [14,15,16], there are a variety of 2D semiconductors used for gas sensing [17], such as molybdenum disulfide (MoS_2_) [18,19], molybdenum diselenide (MoSe_2_) [20], molybdenum ditelluride (MoTe_2_) [21], tungsten disulfide (WS_2_) [22], tungsten diselenide (WSe_2_) [23], and germanium selenide (GeSe) [24]. Suh et al. showed that a pristine MoS_2_-based sensor at 110 °C has a response to 500 ppm ethanol gas with 25% responsivity [25], and Chen et al. reported that a MoS_2_ gas sensor detected 40 ppm ethanol, acetone, toluene, and hexane gases with responses of 11%, 12%, 6%, and 4%, respectively [26].

While achieving high sensitivity is important in gas sensor development, the primary challenge is to improve the selectivity of sensors to specific gases in a mixture of gases [27]. To improve selectivity, there are several proposals for complex active materials or sensor structures. For example, Yan et al. investigated the selectivity of ethanol gas on a MoS_2_ nanosheet decorated with SnO_2_ nanoparticles. They reported that the response of the sensor to 500 ppm ethanol gas at 280 °C is 160%, which is higher than its response to ammonia, formaldehyde, and acetone gases [28]. Chen et al. investigated the selectivity of a MoS_2_ gas sensor decorated with Au nanoparticles to enhance the sensor response from 12% to 32% compared to the pristine MoS_2_ for acetone gas [26]. Sing and Sharma demonstrated the high responsivity of a MoS_2_/WO_3_ composite structure for hydrogen sulfide and ethanol gases and reported a gas response of 17% for 5 ppm ethanol gas at 260 °C and 15% for 0.5 ppm hydrogen sulfide gas at 320 °C [29]. Verma and Yadav showed good selectivity with a WO_3_/WS_2_ heterostructure gas sensor for acetone gas at 132.5 and 17 at 1 and 1000 ppm concentrations compared to other gases such as ethanol, ammonia, formaldehyde, methanol, toluene, water, xylene, and ethylene glycol [30]. Hussain et al. investigated the selectivity of WSe_2_ nanosheets and WSe_2_/reduced graphene oxide hybrid sensors and reported that the hybrid sensor has a response of 5.5% for 100 ppm ethanol at 180 °C while the WSe_2_ sensor has a 2.75 times lower response. Additionally, they investigated the sensors’ selectivity for ethanol compared to other gases such as methanol, toluene, isopropyl alcohol, ammonia, and acetone [31]. Nanorod-decorated nanosheets, including vanadium oxide (V_2_O_5_) and samarium (III) oxide (Sm_2_O_3_), have also been employed in toluene sensors as well as acetone sensors and showed good selectivity and sensitivity [32,33].

In this study, instead of using complex or hybrid active material systems, we propose a dual-channel MoS_2_ gas sensor for VOCs consisting of two sensors in the same active material by modifying half of the MoS_2_ nanosheet via a simple step of ultraviolet–ozone (UV-O_3_) exposure. Therefore, the device is composed of a pristine and a UV-O_3_-treated MoS_2_ sensing element on the same chip. In our previous study [34], we have shown that an appropriate UV-O_3_ treatment causes a p-type doping effect in the MoS_2_ as a result of the substitution and adsorption of oxygen radicals (O, O_2_, and O_3_). The UV-O_3_ treatment in this study will provide a simple mechanism to regulate the free electron density in the active sensing material, leading to an improved selectivity for polar gas molecules. Sensor fabrication and characterization techniques are presented in the experimental section. The sensing performance and different mechanisms between pristine and UV-O_3_-treated sensors are discussed in the results and discussion section.

## 2. Experimental Details

### 2.1. Surface Characterization

Bulk single-crystal MoS_2_ was purchased from HQ Graphene. The Scotch tape mechanical exfoliation method was used to obtain few-layer MoS_2_ flakes. We first pressed the Scotch tape repeatedly onto the bulk material, and then peeled it off slowly. The process was repeated with fresh tape until flakes of the desired thickness were obtained. We then placed the flakes on clean glass substrates to perform surface characterizations before and after UV-O_3_ treatment. The water contact angle (WCA) is a crucial parameter used to characterize the wettability of a surface by water. We employed the sessile drop method to measure the WCA of the clean Borofloat^®^ glass (PI-KEM, UK) surface and the MoS_2_ flakes surface at a macroscopic level. As the size of a single flake is limited, we prepared a sample with a high density of few-layer MoS_2_ flakes covering an area much larger than the water droplet (2 microliters). A high-resolution camera was used to capture the images of the droplet profile. We measured the WCAs before and after 4 min UV-O_3_ treatment.

Electrostatic force microscopy (EFM imaging was performed using an atomic force microscope in tapping mode (XE100, Park System, Suwon, Republic of Korea) with a conductive tip (NSC14/Cr-Au). The tip bias was set to 1 V and the electrostatic-force-induced phase-shift signal was acquired at a height of 10 nm from the surface.

### 2.2. Device Fabrication

The Scotch tape and polydimethylsiloxane (PDMS)-assisted mechanical exfoliation method was used to obtain few-layer MoS_2_ flakes. Using a microscope, we identified flakes with the desired number of layers on PDMS (Gel-pak, Hayward, CA, USA). The flake was then transferred from PDMS to a SiO_2_ (300 nm)-on-Si substrate with pre-patterned electrodes. The post-transfer sample was then coated with a resist layer of polymethyl methacrylate (PMMA, Allresist GmbH, Strasberg, Germany) to allow a window to be opened using the electron beam lithography process (EBL, FIE Versa3D Dual Beam, Hillsboro, OR, USA). Half of the 5-layer flake was exposed to air. This area of the MoS_2_ nanosheet was exposed to UV-O_3_ (Ossila, Sheffield, UK) for 4 min. The remaining PMMA resist was then removed in acetone. The electrode fingers were patterned on the sample by EBL with a bilayer resist, MMA/PMMA (Allresist GmbH, Strasberg, Germany), and Au/Cr (60 nm/10 nm) were deposited as the contact electrodes. A lift-off process was carried out in warm acetone. The fabricated sensor was mounted onto a ceramic chip holder with wires bonded for electrical characterization.

### 2.3. Photoluminescence and Raman Spectroscopy Measurements

Micro-photoluminescence (micro-PL) and micro-Raman measurements were carried out using a custom-built micro spectroscopy set-up, equipped with a thermoelectric cooled CCD (Newton BEX2-DD, Andor, Oxford Instruments, UK) and two gratings (300 and 1800 grooves/mm) in a spectrometer (Shamrock 500i, Andor). To optically excite the samples, a 532 nm CW laser (Gem532, Novanta Photonics, Taunton, UK) was used, and the excitation laser beam was focused to a spot that was ~0.8 μm in diameter on the samples placed on an XYZ translation stage via a 100x NIR objective (NA = 0.7). Spectra from the samples were collected via the same objective.

### 2.4. Gas-Sensing Setup

The VOC vapor was generated from cooled bubblers that were immersed in a thermally controlled bath, with synthetic air as the carrier gas. The gas stream saturated with the VOC analyte was then diluted with pure synthetic air to achieve the desired gas concentration. The total gas flow rate was set to 300 sccm by the computer-controlled mass flow controllers. The current–time characteristics of the gas sensors were measured under a constant bias of 1 V using a KEITHLEY 6517B Electrometer. A LakeShore 340 (Lake Shore Cryotronics, Westerville, OH, USA) temperature controller was used to control the ambient temperature. All the gas-sensing experiments were carried out by introducing the VOC target gas mixed with dry air for 10 min followed by a flow of pure dry air for 10 min in each cycle. Gas-sensing experiments were carried out by monitoring the current change of the sensors with a constant voltage supply (1 V). The response of the sensor was calculated based on the resistivity changes of the active material so that it could be compared independently of the device geometry (with different channel widths and lengths). The response is defined as *R* = $\left(\frac{\rho_{i}\;-\;\rho_{s}}{\rho_{i}}\right)*100\%,$ where $\rho_{i}$ and $\rho_{s}$ represent the resistivities of the gas before and after contact with the sensor, respectively. The resistivity of the sensor is defined as $\rho =\left(R*w\right)/l\;$, where *R*, *w*, and *l* represent resistance, contact width, and channel length, respectively [35].

## 3. Results and Discussion

### 3.1. Surface Characterization

We carried out WCA measurements and EFM imaging to characterize the surface of few-layer MoS_2_ before and after the UV-O_3_ treatment. Before the UV-O_3_ treatment, we observed that the WCAs are very different between a blank substrate (Borofloat^®^ glass, 29° ± 2°) and a glass substrate covered with MoS_2_ flakes (78° ± 15°), as shown in Figure 1a,c, indicating that the blank glass substrate is much more hydrophilic than the MoS_2_ surface, which is consistent with the literature [36]. The large uncertainty in the WCA values originated from the variation in flake density over a large area of the sample. After a 4 min UV-O_3_ treatment, the WCA increased by 15% for the MoS_2_ sample (becoming slightly more hydrophobic) but decreased by 14% for the blank reference sample (slightly more hydrophilic); see Figure 1b,d. UV-O_3_ treatment is known to make glass surfaces more hydrophilic by introducing hydroxyl (OH) groups that attract water molecules. The mild increase in hydrophobicity on the MoS_2_ surface can be attributed to the oxidation of the surface [37,38].

EFM is an electrical mode in AFM to map variations in the sample’s electric field and reveal information about the surface potential and charge distribution. The surface potential is obtained relative to the tip potential. In our EFM measurements, we kept the tip potential and the distance from the tip to the surface constant and obtained surface potential mapping and the average surface potential from the active region of the sensing devices. The average surface potential changes from 0.094 mV to −112.5 mV after a 4 min UV-O_3_ treatment; see Figure 1e,f. This reduction in the surface potential is due to the reduction in the Fermi level of the MoS_2_ flake, resulting in p-type characteristics [39].

### 3.2. Photoluminescence and Raman Spectroscopy of MoS_2_

The layer number of the flake was further confirmed based on the peak wavelength of the PL spectrum. A PL peak wavelength at 890 nm indicates that the flake is a five-layer (5 L) MoS_2_, see the highlighted area in Figure 2a (I). In our previous study, we observed that 4 min of UV-O_3_ exposure time was optimal to produce p-type-doped MoS_2_ while maintaining above 80% of the initial PL intensity [34]. Therefore, half of the active material was treated for 4 min while the other side was kept pristine under the resist, see Figure 2a (IV), in order to create two channels in the same material for sensing. Figure 2b,c show the PL and Raman spectra of the pristine and the UV-O_3_-treated MoS_2_, respectively. The PL spectra of two sides of the material were used as a control indicator of whether the UV-O_3_ treatment degraded the material or not. With maintaining 70% of the initial PL intensity, it can be concluded there is no structural degradation in the 4 min treated material. On the other hand, the comparison of the Raman spectra between the pristine and treated materials was used as an indicator if the treated material indeed turned into being p-type-doped. The Raman spectrum of MoS_2_ is characterized by two main vibration modes, E_2g_ and A_1g_, representing the in-plane and out-of-plane vibrations, respectively [40]. It is known that the E_2g_ mode is sensitive to the strain while the A_1g_ vibration mode is sensitive to the doping effect [40,41,42]. Kang et al. and Shi et al., in different studies, showed that changes in the line width and blueshifts (toward the higher frequency) of the A_1g_ vibration mode are indications of the p-type doping effect in MoS_2_ [41,43]. The Raman spectrum in Figure 2c shows 0.8 cm^−1^ of blueshift in the A_1g_ vibration mode, which can be attributed to the p-type doping effect in MoS_2_. After the PL and Raman examinations, the electrodes were fabricated to achieve two sensing channels in the same device.

### 3.3. Sensing Performance

The current–time (I-t) characteristics were investigated to examine the saturation time, i.e., time before the current saturates with the applied voltage. Figure 3a,b show the room-temperature I-t characteristics of pristine and treated sensors, respectively. The p-type doping effect due to the UV-O_3_ treatment results in reduced contact resistance, and hence the measured current increases 50 times after the treatment. Although there is a difference in the magnitude of the current for these two sensing channels, the current saturates after 5 min of applied voltage in both channels. This saturation time was taken into consideration in the sensing experiments.

Gas measurements were carried out with the following flow sequence, with the gas flow system illustrated in Figure 3c: 10 min flow of dry air, followed by 10 min flow of target gas mixed with dry air, and then 10 min flow of dry air followed by a flow of a different concentration of gas mixture. In this study, we aimed to show how post-growth functionalization of 2D materials can affect their gas-sensing performance. To understand the mechanism behind the change in performance, we eliminated all other environmental/external effects, such as 2D material layer thicknesses and humidity [44]. We examined the sensing performance at room temperature (RT) and 100 °C. The response of the sensors for each gas was measured with seven different concentrations in the range of 300 ppm and 15,000 ppm. The individual range was chosen based on their occupational environment gas exposure limit. Figure 4 shows the real-time monitoring of the current change of the pristine MoS_2_ gas sensor with varying gas concentrations at RT and 100 °C. While there is no response in the sensors for VOC gases at RT, reasonable responses were observed in the pristine sensor to ethanol, acetone, and toluene at 100 °C.

The pristine MoS_2_ used in this study is intrinsically n-doped. The main gas-sensing mechanism of the pristine MoS_2_ sensors is the same as any n-type bulk semiconductor-based sensor. In an n-type semiconductor where electrons are the majority charge carriers, the oxygen molecules in the air are physically adsorbed, by trapping free electrons, onto the surface, resulting in space charge layers that are called electron depletion layers (EDLs). Because the physically bonded oxygen reacts with sensing molecules and forms chemical bonds, the resistance of the sensor reduces. The types of chemisorbed ionized-oxygen species (O^−^, O_2_^−^, or O^2−^) are largely affected by the working temperature as seen in Equations (1)–(4) [45].
O_2(gas)_→O_2(ads)_(1)
O_2(ads)_ + e^−^ → O_2_^−^
_(ads)_       (<150 °C) (2)
O_2_^−^ _(ads)_ + e^−^ → 2O^−^
_(ads)_      (150 °C–400 °C) (3)
O^−^
_(ads)_ + e^−^ → O^2−^
_(ads)_      (>400 °C) (4)

When a pristine MoS_2_ sensor is operated at 100 °C, the O_2_^−^ ion plays a more important role in detecting VOCs. The sensing mechanism in the pristine MoS_2_ sensor is illustrated in Figure 5.

Ethanol, acetone, and toluene gases react with surface oxygen ions. The interaction mechanism of the chemical reactions at 100 °C is given by Equations (5)–(7) for ethanol, acetone, and toluene, respectively.
C_2_H_5_OH_(g)_ + 3O_2_^−^
_(ads)_ →3H_2_O_(g)_ + 2CO_2(g)_ + 3e^−^(5)
CH_3_COCH_3(g)_ + 4O_2_^−^
_(ads)_ →3H_2_O_(g)_ + 2CO_2(g)_ + 4e^−^(6)
C_7_H_8(g)_ + 9O_2_^−^ _(ads)_ → 7CO_2(g)_ + 4H_2_O_(g)_ + 9e^−^(7)

As a result of these reactions, different amounts of CO_2_, H_2_O, and numbers of electrons are generated with the presence of ethanol, acetone, and toluene, respectively. The removal of O_2_^−^ ions from the surfaces due to reactions with the target gas, for example, ethanol, causes a thinner EDL region and a decrement in the potential barrier, which results in an increment in the sensor current as shown in Figure 6. The trend of different responses to different gases follows the amount of required O_2_^−^ for each gas molecule; see Equations (5)–(7). Therefore, we can conclude that the response of the sensors was dependent on the thickness of EDLs [11,46].

Figure 6 shows the real-time monitoring of the current change in a few-layer UV-O_3_-treated MoS_2_ sensor with varying gas concentrations at RT and 100 °C. Reasonable responses are observed for ethanol and acetone at 100 °C compared to no notable response at RT. The limits of detection (LODs) are estimated to be <300 ppm, <1500 ppm, and <500 ppm for ethanol, acetone, and toluene, respectively. Characterizing the dual-channel sensors at a wider range of temperatures may offer further improvement in sensitivity and selectivity, which can be investigated in future work. The responses of the pristine few-layer MoS₂ sensors to 3000 ppm ethanol, acetone, and toluene gases at 100 °C are 18%, 3.5%, and 49%, respectively. The UV-O_3_-treated MoS₂ sensors show responses of 13.4%, 3.1%, and 6.7%, respectively. Therefore, such a dual-channel sensing device demonstrates a 7-fold improvement in selectivity for toluene against ethanol and acetone. Due to the toluene molecules requiring two to three times more O_2_^−^ _(ads)_ ions to react with compared to the ethanol and acetone molecules (see Equations (5)–(7)), the response to toluene gas is expected to be affected much more significantly by the different EDL layers formed on top of the active surface before and after the UV-O_3_ treatment compared to the other two gases, resulting in an enhanced selectivity for toluene.

It has been revealed that UV-O_3_ treatment leads to the formation of S-O bonds in the MoS_2_. The O^−^ ions can also exist at interstitial sites and S-vacancies [47]. The physical and oxygen-mediated chemical bonds in the UV-O_3_-treated sensors result in a p-type characteristic in the MoS_2_. Therefore, UV-O_3_-treated sensors showed a reduced response to toluene compared to acetone and ethanol due to its large molecular structure and more ionized oxygen required to bind (Equation (7)). Figure 7 shows the comparison of the responses for the pristine and UV-O_3_-treated MoS_2_ sensors. The response of the UV-O_3_-treated sensor reduces for all VOCs except acetone at higher concentrations. This reduction is attributed to switching the type of the free carriers from electron-dominant to hole-dominant, resulting in a reduction in the dynamic EBL thickness of the sensor surface during the operation. To achieve a better resolution and higher sensitivity, a larger number of sensing channels can be pursued, inspired by the structure proposed by Rabchinskii et. al. [16], with large area flakes or wafer-scale MoS_2_ grown by chemical vapour deposition. Furthermore, the layer-thickness-dependent sensitivity can be explored in future studies.

## 4. Conclusions

We have designed and fabricated a few-layer MoS_2_-based dual-channel gas sensor platform for volatile organic compounds, including ethanol, acetone, and toluene. The multi-pixelated sensor platform consists of two sensors side by side made from the same material. While one-half of the material is kept as pristine, i.e., an electron-dominant sensor, the other half of the material is treated by UV-O_3_ to obtain a hole-dominant sensor. UV-O_3_ treatment is used to control the free electron density in the active material and change the thickness of the electron depletion layers. Other techniques such as oxygen plasma or annealing in air or an O_2_ environment can also be employed for the same purpose, but we believe that UV-O_3_ treatment is more reliable and reproducible. The responses of pristine MoS_2_ gas sensors to 3000 ppm of ethanol, acetone, and toluene gases were found to be 18%, 3.5%, and 49%, and those of UV-O_3_-treated sensors were 13.4%, 3.1%, and 6.7%, respectively, with 0% humidity at 100 °C. The proposed double-channel sensor system reveals excellent selectivity to toluene, which can be further integrated in future work.

## Figures and Tables

**Figure 1 nanomaterials-14-00633-f001:**
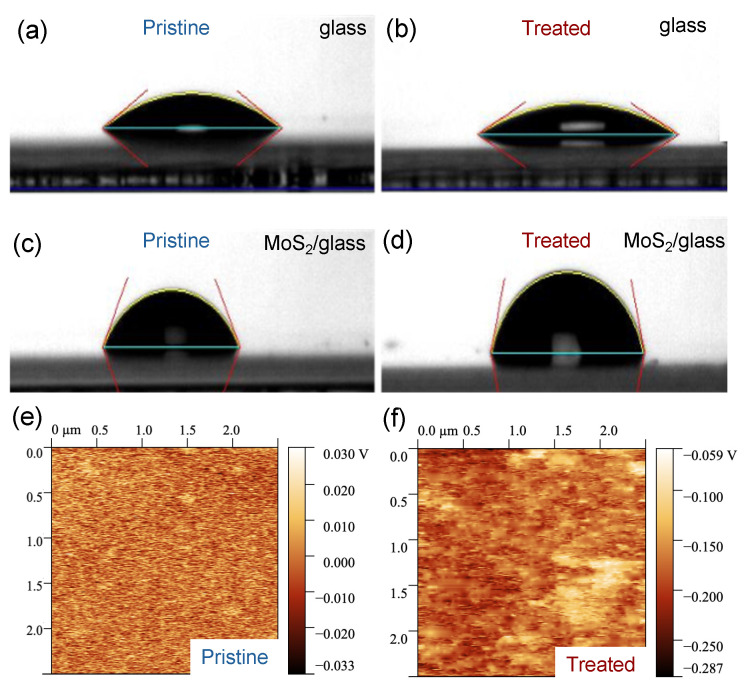
Surface characterization before and after UV-O_3_ treatment. (**a**–**d**) The water contact angles (WCA) were measured on a blank Borofloat^®^ glass and a Borofloat^®^ glass with a high density of few-layer MoS_2_ flakes before (**a**,**c**) and after (**b**,**d**) a 4 min UV-O_3_ treatment, respectively; (**e**,**f**) are the electrostatic force microscopy (EFM) images of the 5 L MoS_2_ flake surface before and after UV-O_3_ treatment, respectively.

**Figure 2 nanomaterials-14-00633-f002:**
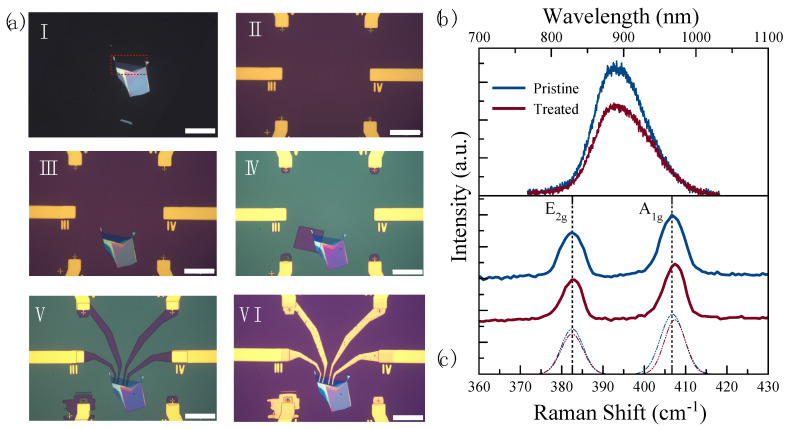
(**a**) Optical microscope images (with a scale bar of 50 μm) of the sensor in different fabrication steps: I. As-exfoliated few-layer MoS_2_ flake on polydimethylsiloxane (PDMS) with the 5-layer area highlighted with red dotted lines; II. Contact electrodes fabricated on SiO_2_-on-Si substrate with a 200 μm× 200 μm clear window for transferring 2D materials; III. After transferring the few-layer MoS_2_ flake onto SiO_2_-on-Si substrate; IV. After opening a window in the PMMA resist layer on half of the 5-layer MoS_2_ for the UV-O_3_ treatment, V. After patterning the electrode fingers to connect the MoS_2_ flake and the prefabricated contact pads in the MMA/PMMA resist; VI. Final device after metallization and lift-off process; (**b**) PL and (**c**) Raman spectra of the pristine (blue) and the UV-O_3_-treated (red) MoS_2_.

**Figure 3 nanomaterials-14-00633-f003:**
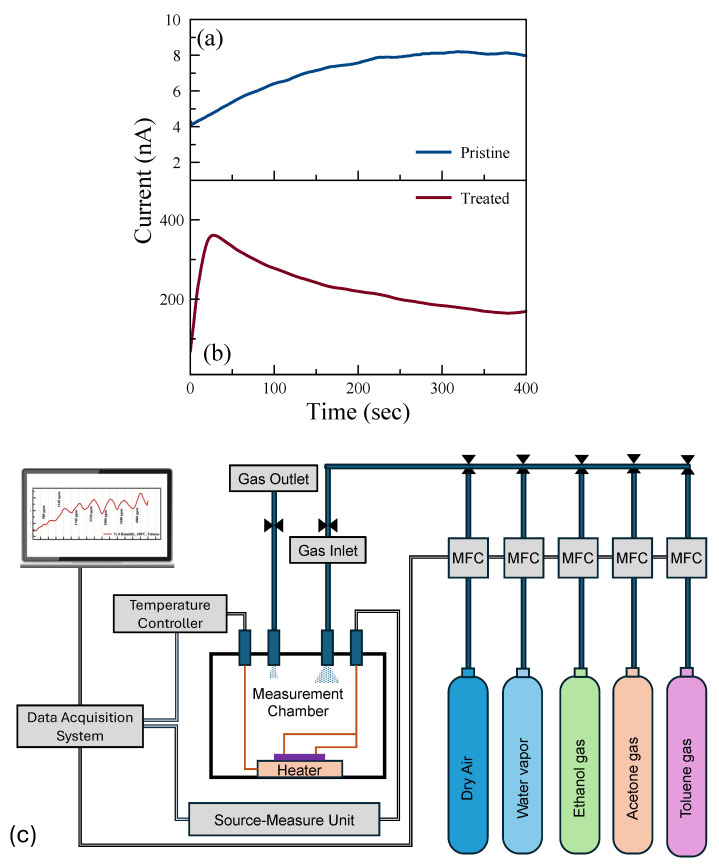
Current–time (with a constant bias of 1 V applied) plots of (**a**) pristine and (**b**) UV-O_3_-treated MoS_2_, respectively, and (**c**) a schematic illustration of the gas measurement system.

**Figure 4 nanomaterials-14-00633-f004:**
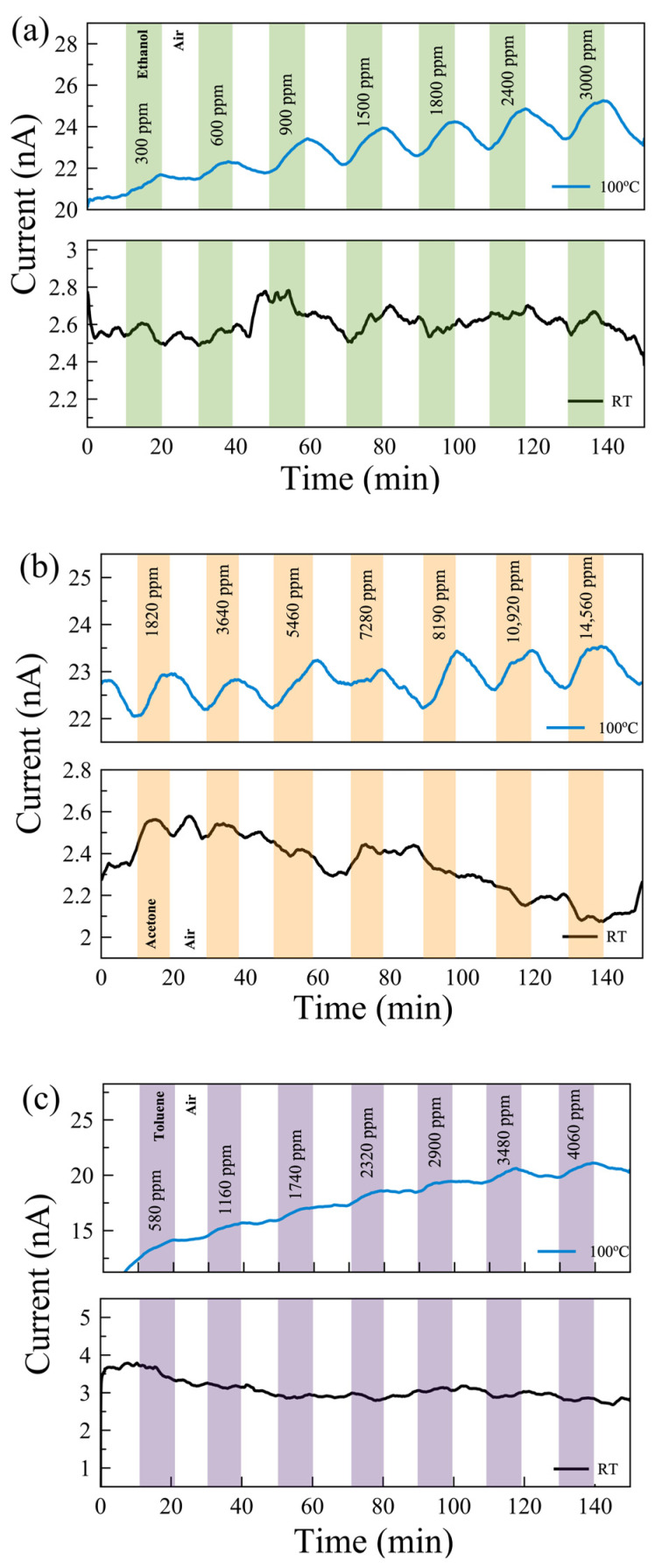
Real-time monitoring of current change in a few-layer pristine MoS_2_-based gas sensor with varying gas concentrations of (**a**) ethanol, (**b**) acetone, and (**c**) toluene gases at 100 °C (top) and room temperature (RT, bottom).

**Figure 5 nanomaterials-14-00633-f005:**
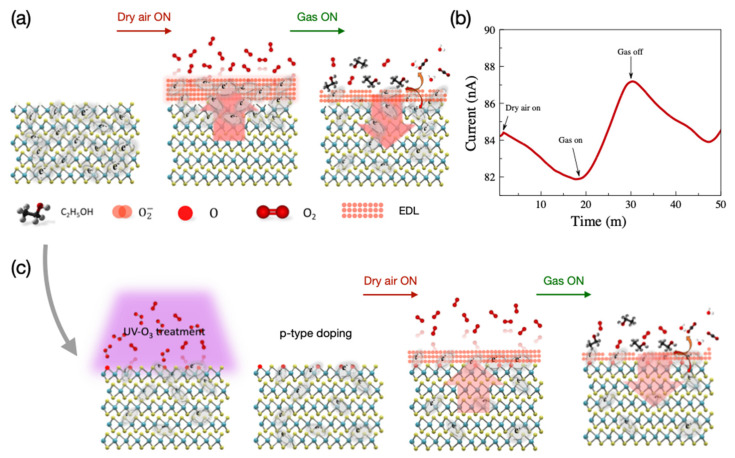
(**a**) Illustration of the sensing mechanism, (**b**) response of the pristine MoS_2_-based gas sensor to dry air and ethanol gas at 100 °C, and (**c**) illustration of the sensing mechanism of UV-O_3_-treated sensor.

**Figure 6 nanomaterials-14-00633-f006:**
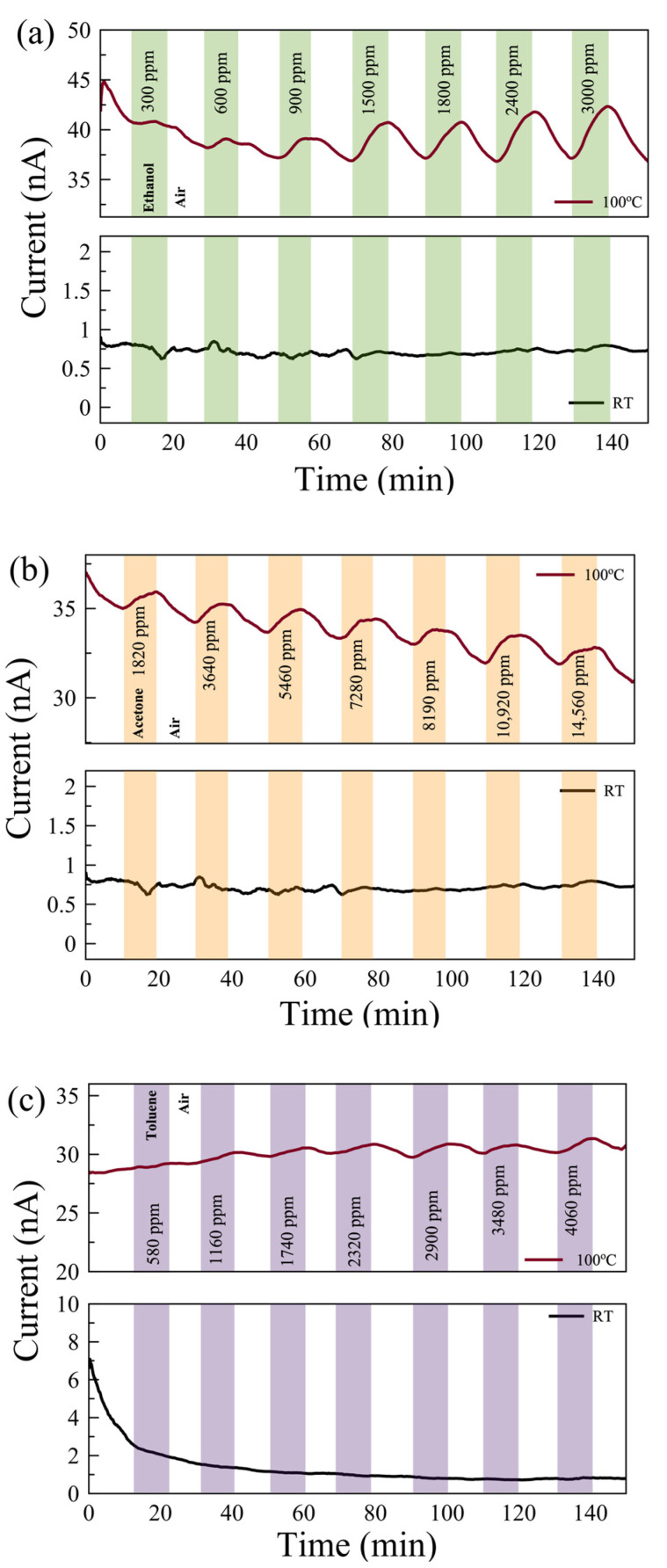
The responses of UV-O_3_-treated few-layer MoS_2_ gas sensors exposed to (**a**) ethanol, (**b**) acetone, and (**c**) toluene gases at 100 °C (top) and at RT (bottom). The background drifts in the current values are due to the carriers stabilizing in the sensor.

**Figure 7 nanomaterials-14-00633-f007:**
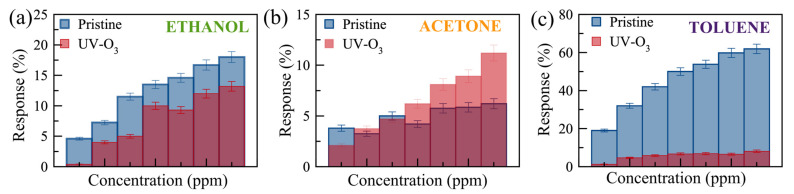
Comparison of the responses to (**a**) ethanol (300 to 3000 ppm, with error bars of 5%), (**b**) acetone (1820–14,560 ppm, with error bars of 8%), and (**c**) toluene (580–4060 ppm, with error bars of 6%) gases for pristine (blue) and UV-O_3_-treated (red) few-layer MoS_2_ gas sensors. The error bars were obtained from the resistivity trends of 4 different devices.

## Data Availability

The authors declare that all the data and code supporting the findings of this study are available within the article, or upon request from the corresponding author.

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
