# Peer review of "A Dual-Channel MoS2-Based Selective Gas Sensor for Volatile Organic Compounds"

_nanomaterials, 2024, doi:10.3390/nano14070633_

Round 1

Reviewer 1 Report

Comments and Suggestions for Authors

The manuscript presents a VOC sensing device using surface-treated dual-channel MoS2. After a thorough review, it is evident that the manuscript requires substantial revisions to meet the journal's publication standards. The article could be acceptable if the major suggestions proposed below are implemented.

1. The authors have introduced several abbreviations including MoS2, MoSe2, MoTe2, WS2, WSe2, GeSe, etc., without prior definitions, despite them being the first occurrence in the manuscript. Please define them upon their initial use.

2. The authors should clearly define how the percent response values are defined and calculated.

3. In the Abstract, it is claimed that the dual-channel sensing device achieves a 7-fold improvement in selectivity for Toluene over Ethanol and Acetone, yet this significant point is not discussed in the main body of the manuscript!

4. The presentation of device fabrication in Section 2.1 is inadequately presented and structured, necessitating significant revisions for improved clarity. The current description falls short of offering a coherent and understandable process, which is crucial for readers aiming to grasp the technical details of your methodology.

5. Please add a schematic illustrating the experimental gas sensing setup.

6. The authors do not provide any rationale for testing the sensor only at room temperature and 100 °C. A wider range of operating temperatures should be explored to better understand the sensor's performance and limitations.

7. The lack of sensor response to the tested VOCs at room temperature raises concerns about the practical applicability of this device. For real-world applications, it is desirable to have sensors that can operate efficiently at room temperature. Please explain this situation.

8. The caption for Figure 2 inaccurately labels the top and bottom curves as representing room temperature and 100 °C, respectively, while the graphs actually depict the opposite, with the top curves corresponding to 100 °C and the bottom curves to room temperature.

9. The authors claim a decreased response to Toluene following UV-O3 treatment, yet Figure 4 suggests a higher sensitivity to Toluene compared to Ethanol and Acetone. Specifically, increasing Ethanol concentrations minimally affect the sensor's response, and surprisingly, increasing concentrations of Acetone seems to somewhat decrease the sensor’s response. Conversely, Toluene's increasing concentrations markedly elevate the sensor's response, indicating superior sensitivity. This observation appears to contradict the claimed selectivity enhancement. Could the authors provide clarification on this apparent discrepancy?

10. It is not stated how many times the experiments were repeated to decide about its important characteristics like sensitivity and specificity!

11. In the main text, the manuscript emphasizes the enhanced selectivity of the UV-O3-treated sensor towards Ethanol and Acetone (lines 197-199). However, both the abstract and conclusion state that the sensor exhibits high selectivity towards Toluene. This discrepancy raises confusion regarding the sensor's actual selectivity. Please resolve this inconsistency and clearly specify the gases towards which the sensor demonstrates higher selectivity.

12. The authors did not investigate the effects of humidity on the sensor's performance. In the experimental section, they mention that "The humidity of the chamber was kept at 0% in order to eliminate the humidity effect on the responses of the sensors. This is a significant limitation of the study, as real-world environments always contain some level of humidity, which can greatly influence the sensor's response to target gases.

Reviewer 2 Report

Comments and Suggestions for Authors

Reviewer report on manuscript nanomaterials-2928214

Kus et al.A dual-channel MoS2-based selective gas sensor for volatile organic compounds

In this work, Authors proposed a novel gas-sensing device that addresses this challenge. It consists of two side-by-side sensors fabricated from the same active material, few-layer MoSâ‚‚, for detecting volatile organic compounds like alcohol, acetone, and toluene. To create a dual-channel sensor, we introduce a simple step into the conventional 2D material sensor fabrication process. This step involves treating one-half of the few-layer MoSâ‚‚ using ultraviolet-ozone (UV-O3) treatment. The responses of pristine few-layer MoSâ‚‚ sensors to 3000 ppm of ethanol, acetone, and toluene gases are 18%, 3.5%, and 49%, respectively.

The manuscript can be accepted after minor revision.  Authors should make the following corrections:

1.      The introduction doesn’t include all relevant references. Some important references in this field should be added, e.g. [Rabchinskii et al. Guiding graphene derivatization for the on-chip multisensory arrays: From the synthesis to the theoretical background // Advanced Materials Technologies, 2022, 7(7), 2101250.] and references there.

2.      The references should be brought up to date 2024.

3.      I recommend Authors to extend “Introduction” and make the comparison between different type of detectors based on 2D materials.

4.      More details to the section 2 “Experimental details” should be added, including information about samples’ fabrication and characterization.

5.      Authors should add to manuscript a small discussion with comparison of their results with previously obtained for other types of detectors, e.g. based on graphene derivatives [Rabchinskii et al. Toward On-Chip Multisensor Arrays for Selective Methanol and Ethanol Detection at Room Temperature: Capitalizing the Graphene Carbonylation // ACS Applied Materials & Interfaces, 2023, 15(23), 28370] and references there.

6. I recommend Authors to plot the data on Figure 1, Figure 2, Figure 3, and Figure 5 in brighter colors and make them more readable.

Reviewer 3 Report

Comments and Suggestions for Authors

See attached file

Comments on the Quality of English Language

 Minor editing of English language required

Round 2

Reviewer 1 Report

Comments and Suggestions for Authors

None

Reviewer 3 Report

Comments and Suggestions for Authors

The authors answered all questions. I recommend their publication in the journal.